# Effect of Normalizing Annealing Temperature on Precipitates and Texture of Nb-Cr-Bearing Decarburized Grain-Oriented Silicon Steels

**Yulong Liu** [1,2], **Chengyi Zhu** [1,2,*], **Juan Jia** [1], **Yong Wang** [1,2], **Yu Liu** [1,2] and **Guangqiang Li** [1,2]

[1]  The State Key Laboratory of Refractories and Metallurgy, Wuhan University of Science and Technology, Wuhan 430081, China; liuyulong@wust.edu.cn (Y.L.); queenyjj@hotmail.com (J.J.); wangyong0911@wust.edu.cn (Y.W.); liuyu629@wust.edu.cn (Y.L.); liguangqiang@wust.edu.cn (G.L.)

[2]  Key Laboratory of Ferrous Metallurgy and Resources Utilization, Ministry of Education, Wuhan University of Science and Technology, Wuhan 430081, China

*  Correspondence: zhchyhsy@wust.edu.cn; Tel./Fax: +86-27-6886-2665

**Abstract:** The evolution of precipitates and texture was investigated in Nb-Cr-bearing decarburized specimens after normalizing at different temperatures. Enough inhibitors, including Nb(C,N), were obtained, of 20~40 nm in size. Increasing normalizing annealing temperature leads to the number density of the precipitates decreasing and that of mean size increasing. The Goss texture content in the decarburized specimens decreases in different degrees compared with the normalized ones. The minimum Goss texture and maximum $\sum 9$ grain boundaries were obtained in the decarburized specimen normalizing at 950 °C and in this specimen, enough fine dispersed inhibitors, weak but relative stable Goss texture and uniform grain size will be beneficial for Goss grains growth in secondary recrystallization, according to the coincidence site lattice theory.

**Keywords:** Nb-Cr-bearing grain-oriented silicon steels; annealing; precipitates; primary recrystallization; Goss texture

---

## 1. Introduction

Fe-3.0%Si grain-oriented silicon steels are widely used as the core materials of electrical transformers due to their good permeability and low core loss [1]. The inhibitors and sharpness of Goss texture in grain-oriented silicon steel have a great effect on its final magnetic properties [2,3]. Since there are some disadvantages of using only AlN(aluminium nitride) or MnS(manganese sulfide) as primary inhibitors [4], AlN or MnS associated with assistant inhibitors are designed to produce grain-oriented silicon steels.

Studies [5–10] have shown that Nb(C,N) (niobium carbonitride) can form by adding Nb to steels. Compared with conventional inhibitors, it has a lower solubility temperature, a smaller precipitate size, and a lower coarsening rate and core loss. Seen from the composition of the steels reported in the above research results, they still adopt AlN or MnS as the main inhibitor. On the other hand, a moderate amount of chromium added to the grain-oriented silicon steels can reduce eddy-current loss. Moreover, adding chromium can make the primary recrystallization microstructure uniform, refine hot rolled plate grains and make the steel have good magnetic properties [4,11]. In addition, compared with the no chromium added sample, the second recrystallization temperature increased by 20 K and the iron loss decreased by about 5% [12]. More Cr content is designed in the present work and the superiority of Cr is expected to be embodied in the grain-oriented steels.

In addition to inhibitors, researchers [13–17] have also found that normalizing annealing treatment could affect the primary recrystallization and result in an excellent performance of the final product.

Reports [14,18] have shown that different annealing temperatures and cooling conditions could make the inhibitors more refined and the texture more uniform. Li et al. [15] obtained the result that normalizing annealing could make silicon steels obtain a sharp Goss texture. But reports on the effect of different normalizing annealing temperatures on the primary crystallization are limited.

In the present paper, the precipitation behavior of the inhibitors and the texture evolution of Nb and Cr synergistically added to grain-oriented silicon steels at different annealing conditions were studied. The optimized normalizing temperature was obtained. The results will elucidate the mechanism of the effect of annealing temperature on precipitation behaviors and the texture evolution of Nb and Cr synergistically added grain-oriented silicon steels, which is expected to provide reference data for developing a proper annealing proposal.

## 2. Materials and Methods

Grain-oriented silicon steel specimens used in the present work were smelt in a 100 kg vacuum induction furnace (Jinzhou Electric Furnace Co., Ltd., Jinzhou, China), According to our experience of simulating steelmaking composition of the samples in moderate scale experiments and industrial production, the smelting temperature was set at 1640 °C, and the liquid steel was cast into 210 mm × 120 mm × (30~50) mm ingots in a fixed copper mold. The chemical composition of the specimens used in this study are listed in Table 1. Since the reheating temperature was about 1400 °C in the conventional grain-oriented silicon steel using MnS and AlN as inhibitors, the ingots in the present experiments were reheated at 1350 °C for 210 min to ensure the inhibitors dissolved completely. The temperatures of the hot rolling process were set by referring to industrial practice. The bloom rolling temperature was 1100 °C and the finishing rolling temperature was 1000 °C. The slab was hot rolled to 2.3 mm and then modeling coiled at 570 °C. Since Nb and Cr are used as assistant elements, two-stage normalizing annealing and single cold rolling with 90% reduction were followed. References [14,15,18] have reported that first stage normalizing annealing temperature varies from 1000 °C to 1120 °C, from 1 min to 6 min, and the second stage annealing temperature changes from 920 °C to 960 °C, for about 3 min. Theoretically, the annealing temperature can be set from 1150 °C to 850 °C, which includes austenite changing to ferrite to obtain more and fine inhibitors. According to the industrial production practice, the first stage annealing temperature was set at 1120 °C and kept for 4 min, and the second stage temperature was set at 920 °C, 950 °C and 980 °C. Then the specimens were air-cooled rapidly to the second stage temperature to anneal at 920 °C, 950 °C and 980 °C for 3 min and cooled in the air, and were named normalized specimen A, normalized specimen B, normalized specimen C. The normalized specimens were cooling rolled to about 0.23 mm and then decarburization annealed at 840 °C for 3 min at the dew-point temperature 45 °C, and which were correspondingly named as decarburized specimen A, decarburized specimen B, decarburized specimen C.

**Table 1.** Chemical composition of the samples/wt.%.

| C | Si | Mn | P | S | Al | Cr | Nb | N | Fe |
|---|---|---|---|---|---|---|---|---|---|
| 0.049 | 3.104 | 0.080 | 0.008 | 0.006 | 0.027 | 0.268 | 0.086 | 0.007 | Balance |

The morphology and chemical composition of precipitates, specimens extracted by carbon extraction-replica technique, were observed and analyzed by transmission electron microscopy (TEM) (Joel, Tokyo, Japan) combined with energy dispersive X-ray spectroscopy (EDS) (Oxford Instruments, Oxford, United Kingdom). The size distribution of precipitates was investigated by TEM. About 100 connected fields were selected from each specimen. The CS-8800 Infrared Carbon Sulphur Analysis Meter (Wuxi jinbo electronic appliance Co., Ltd., Wuxi, China) was used to determine the carbon content of the decarburized specimens. In the present work, the evolution of Goss texture in normalizing and decarburizing bands was investigated, which mainly originates from the subsurface of the hot rolled band. Results such as coincidence site lattice (CSL) grain boundaries, grain size,

especially for the variation of $\sum 9$ grain boundaries around Goss texture, were expected. So, an electron back scatter diffraction (EBSD) system (Oxford Instruments, Oxford, UK) was chosen to analyze microstructure and texture in the present work, which can provide orientation and morphology information. The microstructure and texture along the longitudinal section of the specimens were determined by the electron backscatter diffraction (EBSD) system equipped with a Nova 400 Nano field emission scanning electron microscope (FE-SEM) (FEI Company, Hillsboro, OR, USA). The texture content and grain size were calculated by the software HKL channel 5. In order to ensure the representative of the results, several observation fields were conducted in the experiments. The EBSD results were averaged on the basis of the statistical results.

## 3. Results

### 3.1. Precipitation Characteristics of Inhibitors in the Bands

In Figure 1, most precipitates in the bands are nanoscale rectangle NbN and NbC or Nb(C, N) (in (a,b)), and some precipitates contain AlN (in (c,d)). There are some spherical MnS compounded with NbN or NbC (in (e,f)). Some Nb-bearing precipitates contain CrN (in (g,h)). The surface scan results of Nb-bearing precipitates compounded with MnS and AlN is shown in Figure 2.

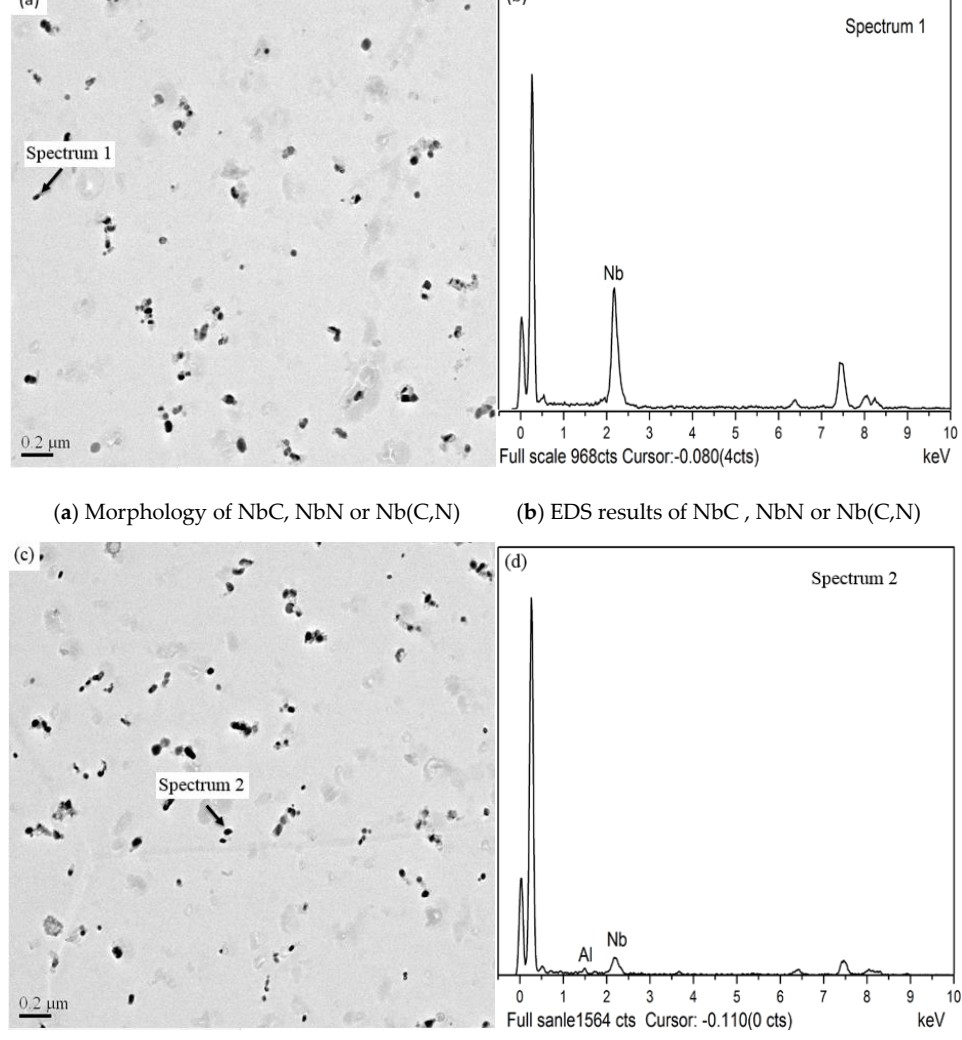

(**a**) Morphology of NbC, NbN or Nb(C,N)					(**b**) EDS results of NbC , NbN or Nb(C,N)

(**c**) Morphology of NbC, NbN or Nb(C,N) and AlN				(**d**) EDS results of NbC, NbN or Nb(C,N) and AlN

**Figure 1.** *Cont.*

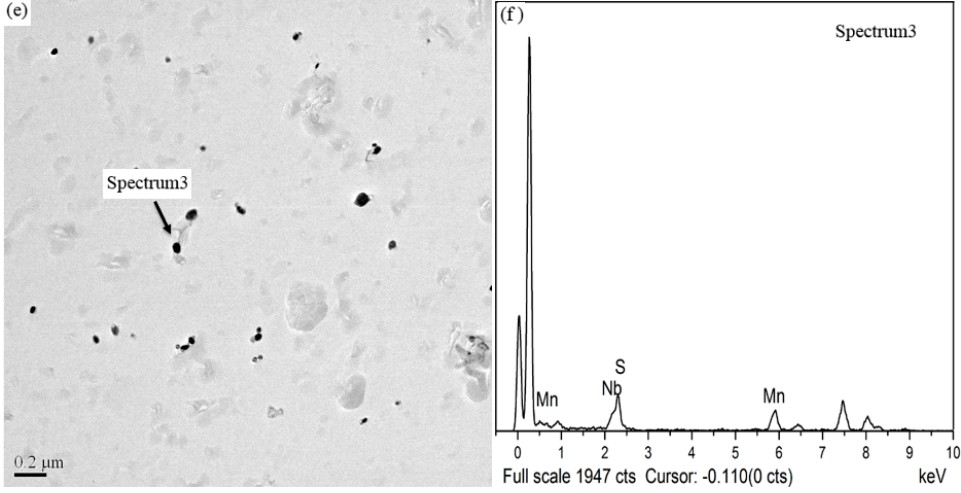

(**e**) Morphology of MnS and NbC, NbN or Nb(C,N)    (**f**) EDS results of MnS and NbC, NbN or Nb(C,N)

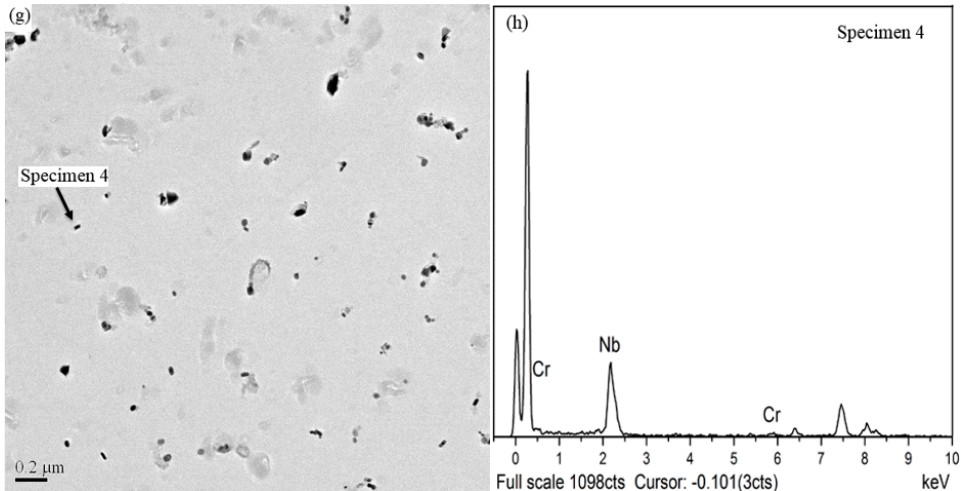

(**g**) Morphology of CrN and NbC, NbN or Nb(C,N)    (**h**) EDS results of CrN and NbC, NbN or Nb(C,N)

**Figure 1.** TEM/EDS results of typical precipitates in the specimens—(**a**–**h**).

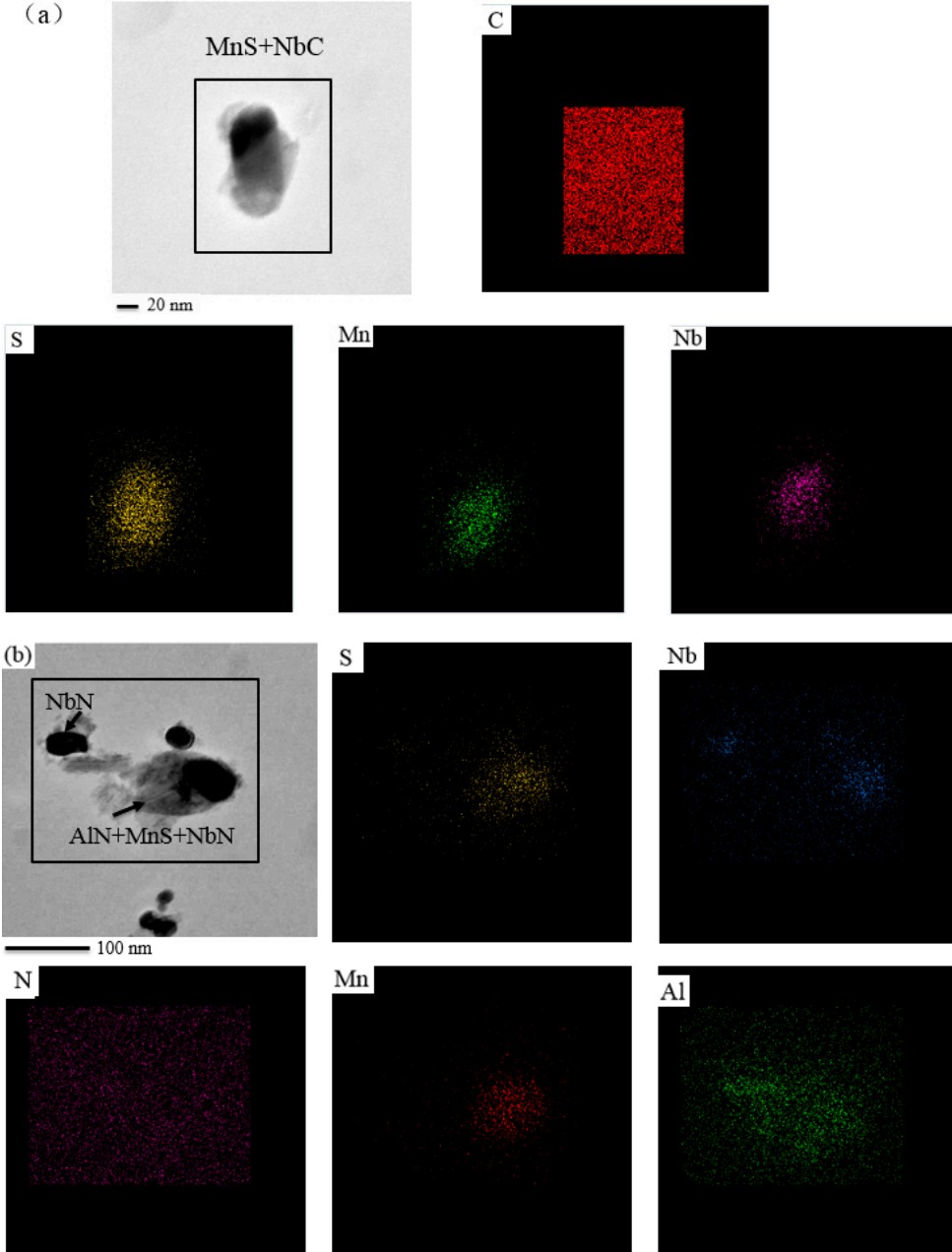

**Figure 2.** Element mapping results of the precipitates.

Figure 3 indicates in normalized specimens, the number density of the precipitates is $12.80 \times 10^6/mm^2$, $3.28 \times 10^6/mm^2$, $2.18 \times 10^6/mm^2$ respectively. After decarbonized annealing, the content of carbon in the specimens was about 0.0054 wt.%. The number density of the precipitates is $8.99 \times 10^6/mm^2$, $6.78 \times 10^6/mm^2$, $3.71 \times 10^6/mm^2$.

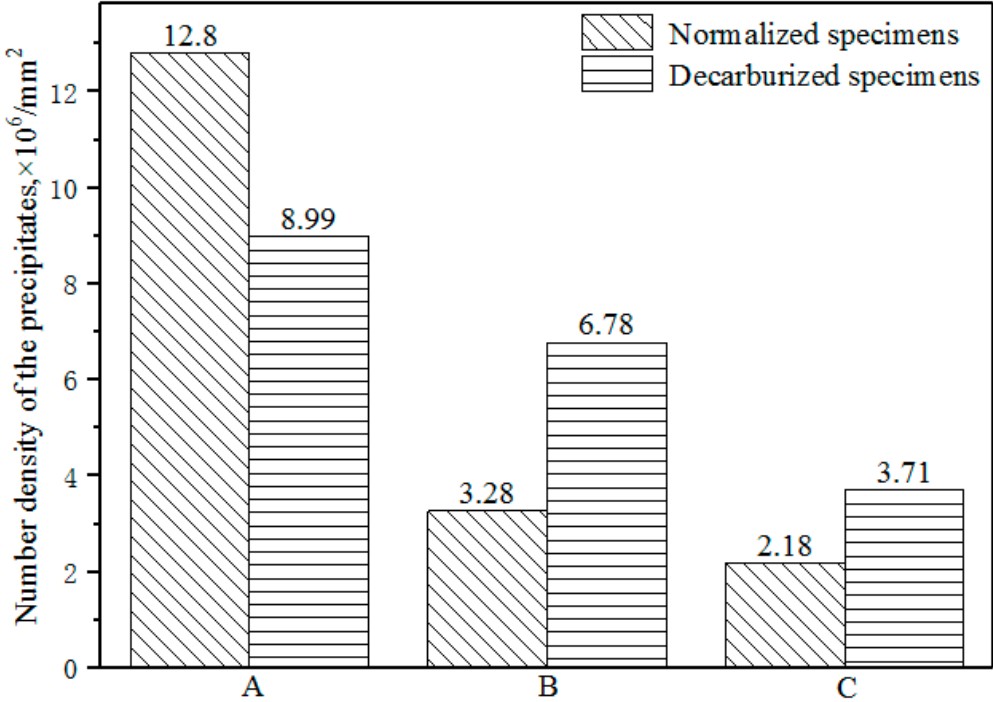

**Figure 3.** Number density of the precipitates in the specimens.

Figure 4a indicates that most size of the precipitates in the normalized specimens is in the range of 20~30 nm. Figure 4b indicates that most size of the precipitates in the decarburized specimens is in the range of 20~40 nm.

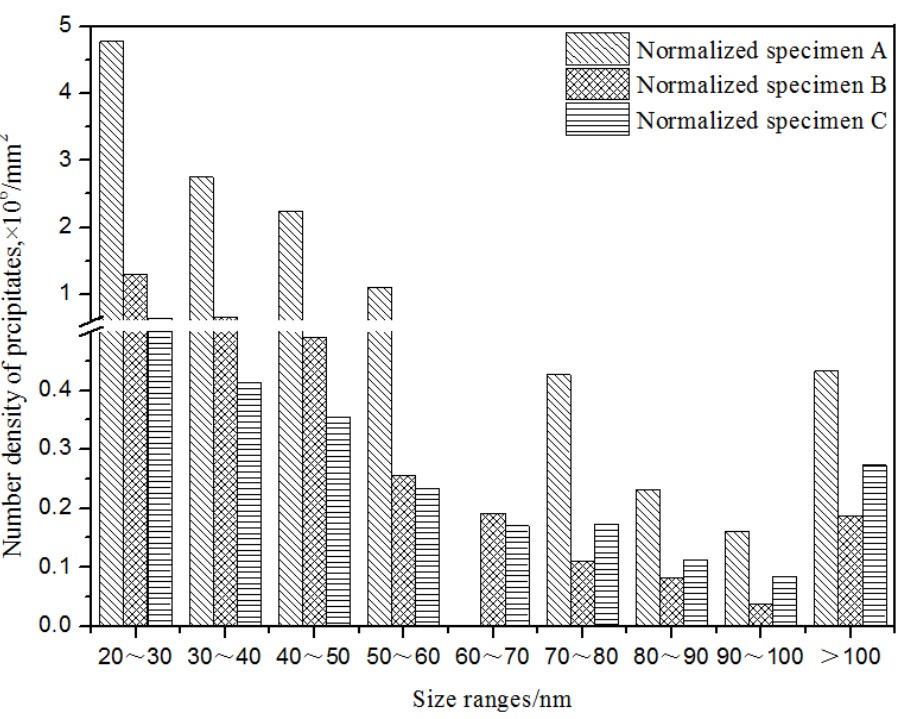

(**a**) In normalized specimens.

**Figure 4.** *Cont.*

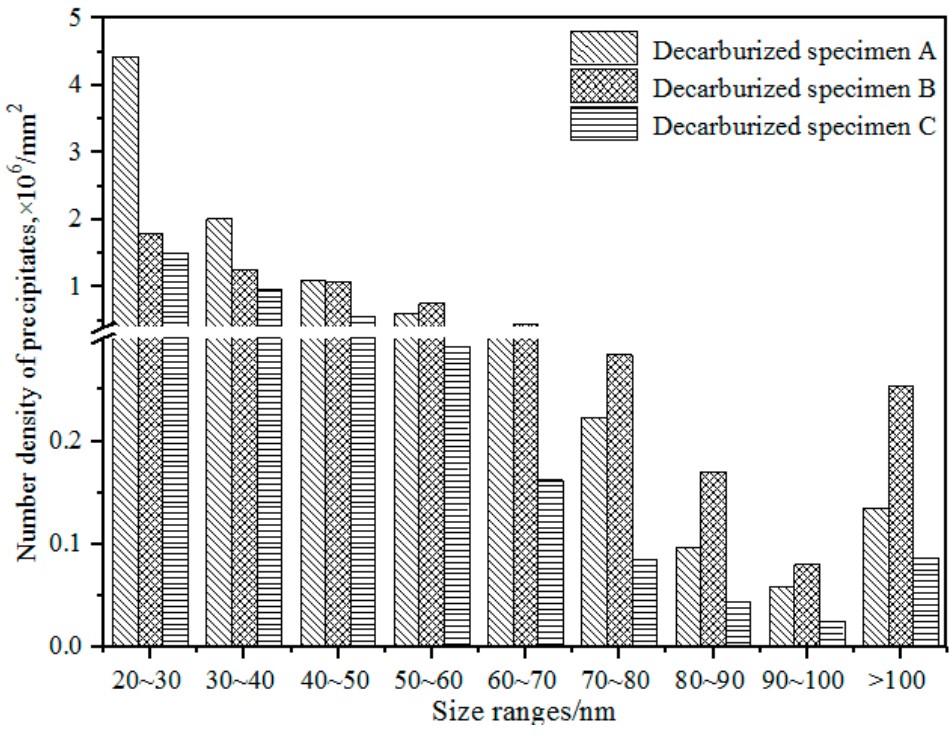

(**b**) In decarburized specimens.

**Figure 4.** Sizes distribution of the precipitates in normalized specimens (**a**) and in decarburized specimens (**b**).

In Figure 5, the mean size is 42.2, 44.4, and 55.4 nm respectively in the normalized specimens. Correspondingly, the mean size is 36.6, 40.1, 39.4 nm respectively in the decarburized specimens.

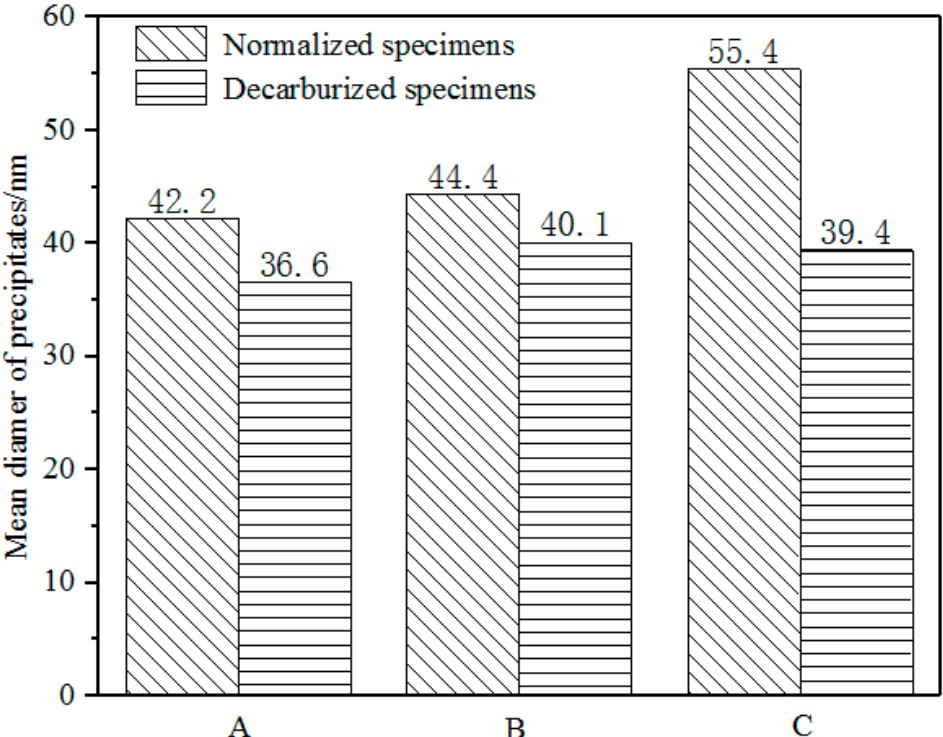

**Figure 5.** Mean size of the precipitates in the normalized and decarburized specimens.

## 3.2. Microstructure and Texture of the Normalized Bands

Figure 6 is the inverse pole figure (IPF) map, which shows the typical microstructure of the normalized bands and shows three regions along the thickness. The surface region is from the boundary to 1/6 thickness of the sample, the subsurface region is 1/6~1/3 thickness and the center region is 1/3~1/2 thickness of the sample (all specimens followed are the same). It can be seen that Goss texture still exists mainly on the subsurface. In addition, the quantity of Goss texture in three specimens is 1.01%, 0.824%, 1.49%.

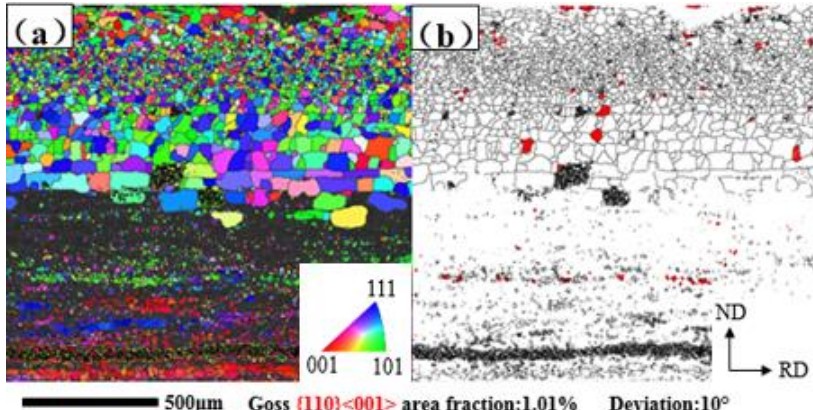

IPF map (**a**) and Goss texture distribution (**b**) of 920 °C normalized specimen.

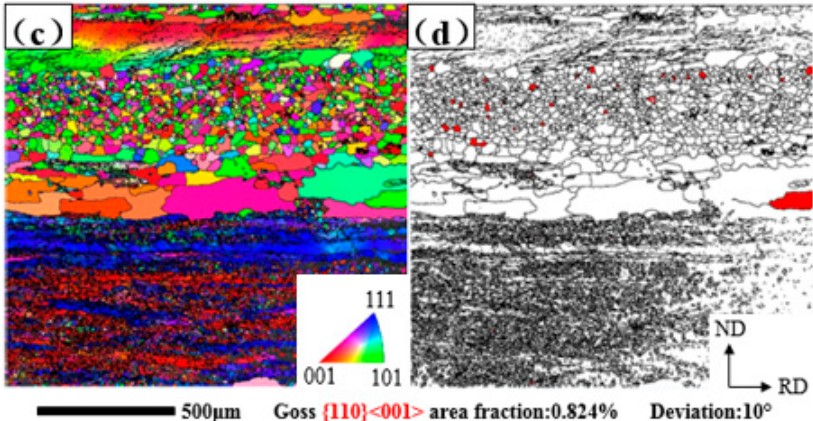

IPF map (**c**) and Goss texture distribution (**d**) of 950 °C normalized specimen.

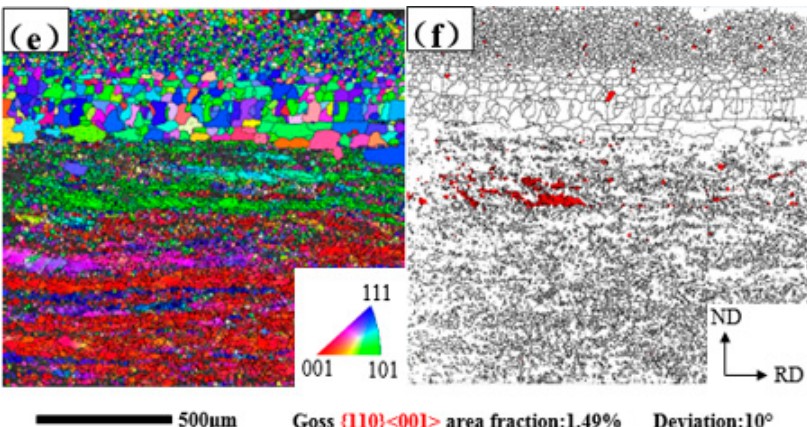

IPF map (**e**) and Goss texture distribution (**f**) of 980 °C normalized specimen.

**Figure 6.** IPF maps of normalizing annealed specimens and the Goss texture distribution.

Figure 7 is the orientation distribution function (ODF) map, which shows the main components of the texture in three samples. In the 920 °C normalized specimen, the texture is {001}<110>, {112}<111>. In the 950 °C normalized specimen, the texture is {111}<110>, {110}<001>. In the 980 °C normalized specimen, the texture is {001}<110>, {111}<110>.

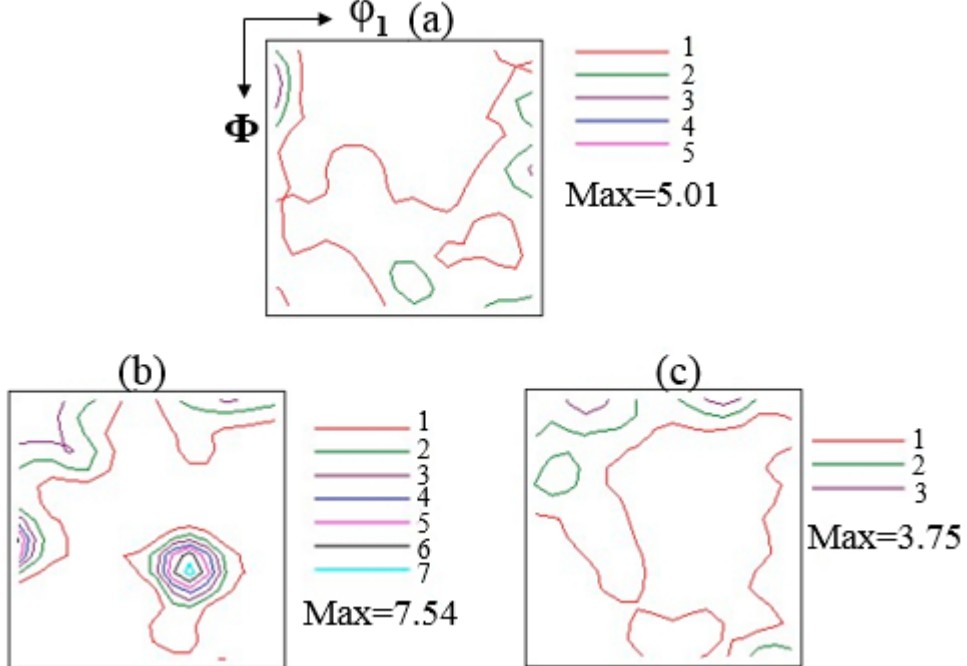

**Figure 7.** Texture ($\phi_2 = 45°$ ODF section) of the specimens at different normalizing temperature, (**a**) 920 °C, (**b**) 950 °C, (**c**) 980 °C.

### 3.3. Microstructure and Texture of the Decarburized Bands

Figure 8 shows the IPF and grain unique color maps of primary recrystallization of the decarburized samples. In Yan's research [19] {111}<112> was the major texture in primary recrystallization specimens. H. Homma et al. [20] concluded that {111}<112>, {411}<148> were related $\sum9$ to Goss oriented grains, which are beneficial for Goss oriented grains to grow up in the following secondary recrystallization process. So, the typical texture in the grain oriented silicon steel chosen to discuss is {111}<112>, {111}<110>, {411}<148>, which is colored in the maps.

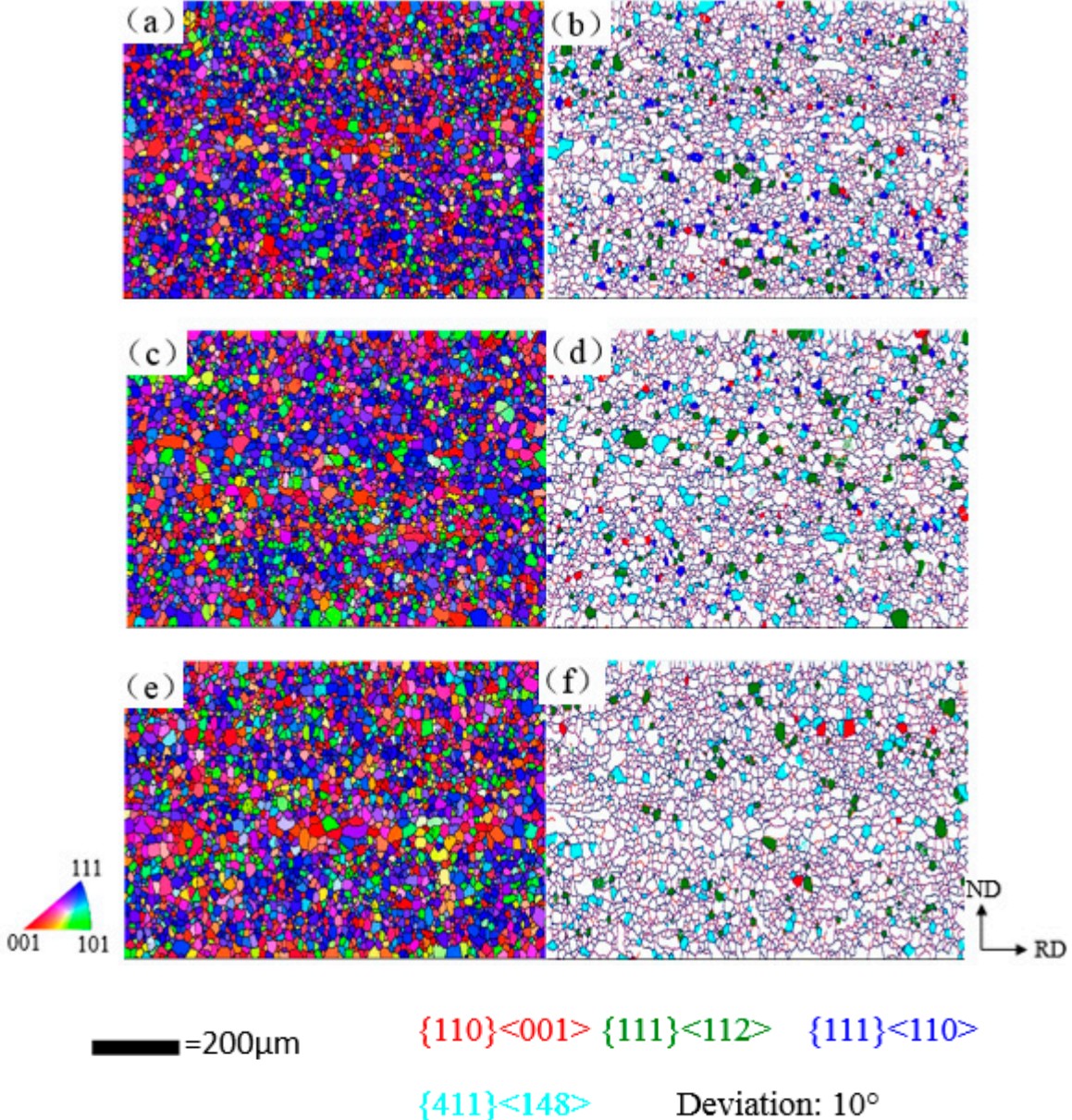

**Figure 8.** IPF maps and main textures distribution of decarburized specimens normalizing. At 920 °C (**a**,**b**), 950 °C (**c**,**d**), 980 °C (**e**,**f**).

In Figure 9, combined with the figures shown in each map with orientation position, the results shown are in accordance with the texture chosen.

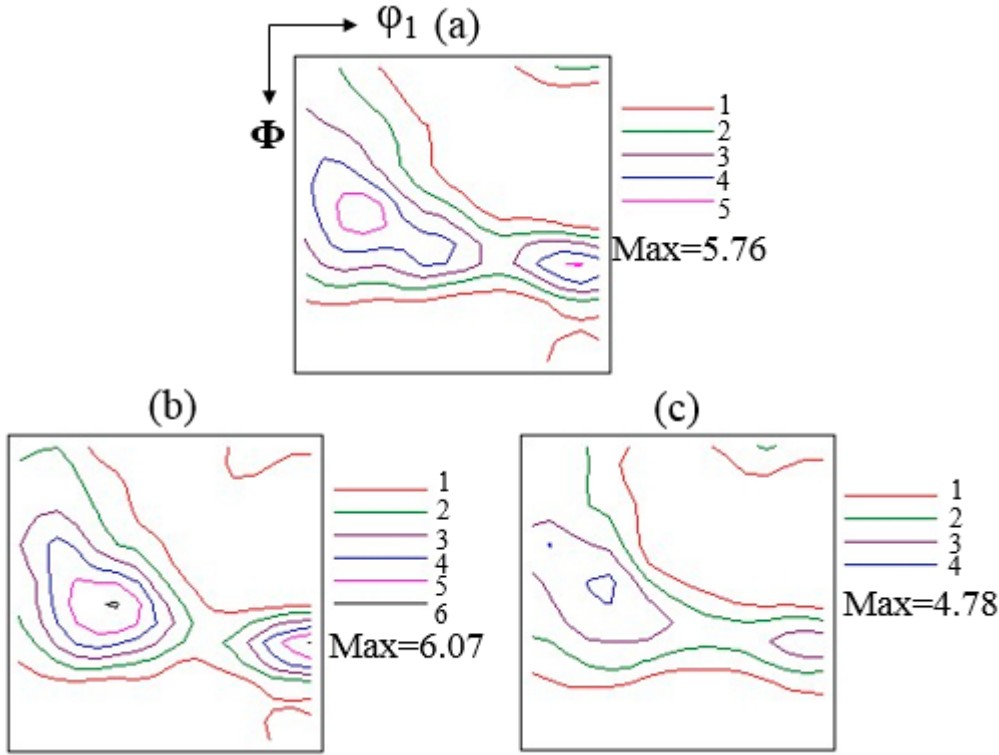

**Figure 9.** Texture ($\phi_2 = 45°$ODF section) of decarburized specimens normalizing annealed at 920 °C (**a**), 950 °C (**b**), 980 °C (**c**).

## 4. Discussion

### 4.1. Effect of Normalizing Temperature on Behaviors of Precipitates in the Bands

EDS results from Figure 1 indicate that the precipitates in the tested specimens are mainly NbC, NbN, Nb(C,N), MnS, AlN or their compounds. $Cr_7C_3$ and CrN are seldom to be found. The beginning-precipitating temperature of the precipitates AlN [21,22], MnS [23,24], NbC [25], NbN [25], $Cr_7C_3$ [26] and CrN [26] in steels can be calculated by Equations (1)–(12).

$$\lg w[\text{Al}]_\alpha \cdot w[\text{N}]_\alpha = -11900/T_\alpha + 3.56 \tag{1}$$

$$\lg w[\text{Al}]_\gamma \cdot w[\text{N}]_\gamma = -7400/T_\gamma + 1.95 \tag{2}$$

$$\lg w[\text{Mn}]_\alpha \cdot w[\text{S}]_\alpha = -10590/T_\alpha + 4.092 \tag{3}$$

$$\lg w[\text{Mn}]_\gamma \cdot w[\text{S}]_\gamma = -14855/T_\gamma + 6.82 \tag{4}$$

$$\lg w[\text{Nb}]_\alpha \cdot w[\text{C}]_\alpha = -8970/T_\alpha + 3.46 \tag{5}$$

$$\lg w[\text{Nb}]_\gamma \cdot w[\text{C}]_\gamma = -5000/T_\gamma + 1.25 \tag{6}$$

$$\lg w[\text{Nb}]_\alpha \cdot w[\text{N}]_\alpha = -13000/T_\alpha + 5.85 \tag{7}$$

$$\lg w[\text{Nb}]_\gamma \cdot w[\text{N}]_\gamma = -10000/T_\gamma + 3.90 \tag{8}$$

$$\lg w[\text{Cr}]_\alpha^7 \cdot w[\text{C}]_\alpha^3 = -25873/T_\alpha + 26.23 \tag{9}$$

$$\lg w[\text{Cr}]_\gamma^7 \cdot w[\text{C}]_\gamma^3 = -22375/T_\gamma + 25.07 \tag{10}$$

$$\lg w[\text{Cr}]_\alpha \cdot w[\text{N}]_\alpha = -8559/T_\alpha + 5.26 \tag{11}$$

$$\lg w[\text{Cr}]_\gamma \cdot w[\text{N}]_\gamma = -6513/T_\gamma + 4.32 \tag{12}$$

where, $w$ represents mass percent concentration of elements in the steel in all equations, wt.%, $T$ is temperature, K, $\alpha$, $\gamma$ represents ferrite and austenite, respectively.

In the normalizing annealing process, the beginning-precipitating temperature of the above inhibitors is calculated respectively according to the present composition of the tested steels. The results are listed in Table 2.

**Table 2.** The beginning-precipitating temperature of the inhibitors in the specimens/°C.

| Inhibitor | AlN | MnS | NbC | NbN | Cr$_7$C$_3$ | CrN |
|---|---|---|---|---|---|---|
| In ferrite ($\alpha$) | 1360 | 1156 | 1023 | 1123 | 485 | 800 |
| In austenite ($\gamma$) | 1031 | 1192 | 789 | 1086 | 405 | 652 |

In austenite, the precipitation order of the precipitates in the tested specimens is MnS > NbN > AlN > NbC > CrN > Cr$_7$C$_3$, while in ferrite, the precipitation order is AlN > MnS > NbC > NbN > CrN > Cr$_7$C$_3$. On the basis of classical nucleation theory, the earlier precipitates will become the nucleus of the later ones. During the first stage of normalizing annealing, some ferrite was transformed to austenite. In addition, when the specimens were air-cooled to 920 °C in the second stage, the "transformed austenite" turned to ferrite again. Due to the solubility production of AlN in austenite is 1~2 orders higher than that in ferrite [21], which changes greatly during the annealing process. The precipitation temperature of NbN is higher than that of AlN in austenite. It was reported that AlN particles have a longer incubation period than MnS particles based on thermodynamic and kinetic calculations [25], while the nucleation of Nb(C,N) in ferrite with a short incubation period was extremely rapid over a wide temperature range [27]. So, Nb(C,N) precipitates preferentially and dominates in the tested specimens.

Besides precipitation start temperature, the available nucleation sites for inhibitors have a significant influence on the precipitation behavior of the steels. Controlling different second stage normalizing temperature, inhibitor forming elements have different diffusion velocity and nucleation sites affected by heat treatment temperature. When the temperature gradient (the difference value between the first stage annealing and the second stage annealing temperatures) is bigger, the available nucleation sites of the inhibitors increases, so the number density of the inhibitors increases (in Figure 3). Furthermore, with the coarsening of precipitated particles, the smaller precipitates decreasing and the bigger ones increasing (in Figure 4a), and the mean size shows a growing trend (in Figure 5)

In the decarburizing annealing process, as the temperature is 845 °C, coarsening may be the main process, the particles range in 20~70 nm decrease slightly, while particles bigger than 100 nm increase slightly (in Figure 4b), so the number density decrease (in Figure 3) and the mean size changes a little (in Figure 5).

The inhibition effect of inhibitors mainly manifests during decarbonization and earlier stage of secondary recrystallization. The number density and size characteristics of inhibitors in the decarbonized specimens inherit that of normalized specimens except at 920 °C. There are so many fine precipitates in the 920 °C normalized specimen, some of them decompose during the decarburization process, which leads to the number density and size decreasing. As a result, fine and homogeneous inhibitors were obtained in the primary annealed specimens, which can provide stronger pinning effect on grain boundary migration. The differences in inhibitor distribution result from the normalizing temperature.

In order to compare the pinning effect of the inhibitors, Zenner force is calculated in the tested specimens. Zenner Equation (13) [28] expresses the relationship between the pinning force and the radius, volume fraction of the fine particles:

$$Z = \frac{3}{4} \cdot \frac{f\sigma}{r} \tag{13}$$

where, $f$ denotes the volume fraction of the second phase particles, $\sigma$ is the interfacial energy, $r$ is the radius of the second phase particles, and $Z$ is the Zenner pinning factor, which is opposed to the driving force of grain boundary migration [29]. The volume fraction of second phase particles can be measured by the McCall-Boyd method [28] can be determined using Equation (14).

$$f = (\frac{1.4\pi}{6}) \cdot (\frac{ND^2}{S}) \tag{14}$$

where, $S$ is the area of measured photos, $D$ is the average diameter of the second phase particles, and $N$ is the number of the second phase particles.

According to Equations (13) and (14), the smaller radius and higher volume fraction of the second phase particles have a great influence on the texture of primary recrystallization. In this paper, through the data in Figure 2, the relationship of pinning forces in three decarburized specimens calculated by Equations (13) and (14) is $Z_{920\,°C} \approx 1.06 Z_{950\,°C} \approx 2.26 Z_{980\,°C}$ (the number and mean size of precipitates are only considered, $Z_{920\,°C}$, $Z_{920\,°C}$, $Z_{920\,°C}$ is the Zenner force of three decarburized specimens respectively). The results indicate that 920 °C or 950 °C normalizing annealing is better for the specimens to obtain larger pining force, which is a benefit for secondary recrystallization.

*4.2. Effect of Normalizing Temperature on Microstructure and Texture of the Annealed Bands*

Normalizing annealing can make the recrystallization grains in subsurface grow, and the grains in the surface coarsen. However, deformed texture in the center of the specimens also recrystallizes, and the size of grains can be calculated by Equation (15) [30]

$$d = K(\frac{G}{\overset{\bullet}{N}})^{\frac{1}{4}} \tag{15}$$

where, $d$ is the mean size of the recrystallized grains, $\overset{\bullet}{N}$ is nucleation rate, $G$ is linear velocity of the grain growth, K is a constant.

Due to the fact strain energy on the subsurface is higher than that in the center of the hot bands, the deformed structure in the subsurface has higher stored energy, which make the subsurface has higher nucleation rate. Meanwhile, the precipitates have pinning effect on annealed grains, which inhibits the annealed grains to grow up. In addition, the change of "ferrite-austenite" in the process of normalizing annealing weakens the Goss texture. That is why the grains size and the content of Goss texture in the normalized bands are different.

With the process of annealing, the precipitates coarsen and make the texture {110}, {100} stronger, but make the texture {111} weaker. The results in Figures 3, 5 and 6 show that the average precipitate size is smaller, the number density of the precipitates and the Goss texture component are larger in the specimens. With the temperature increase, the grains coarsen or carbon decreases, and the number of the grains decreases. In decarburization annealed specimens, the relationship between precipitates and deformed structure recrystallized grains could be explained by Equation (16) [31,32]

$$P = P_D - P_Z = \frac{\alpha\gamma_s}{R} - 3f\frac{\gamma_s}{2r} \tag{16}$$

where $P$ is driving force of sub-crystal growth, $P_D$ is driving force of interface energy, $P_Z$ is pining force of precipitates, $\alpha$ is the coefficient of shape, $R$ and $r$ are respectively for the mean radius of sub-crystal and precipitates, $f$ is the volume fraction of precipitates; $\gamma_s$ is special grain boundary energy.

According to Equation (16), the finer precipitates have stronger restraint effect on the recrystallization of deformed structure. So, the higher quantity and smaller precipitates have more needed time to finish recrystallization. After recrystallization, the precipitates inhibit the growth of recrystallized grains. Therefore, combined with Figures 3 and 5, it is why the mean grain size of the decarburized specimen is 8.52, 9.16, 9.06 μm respectively. In Nakashima's [33] research, excessive cold

rolling deformation can make the most readily available Goss texture change to other texture at about 80% deformation, thus reducing the content and strength of Goss texture in annealed sheets. In this paper, the cold reduction was about 90%, the contents of Goss texture in three specimens were 0.77%, 0.75%, 0.78%, all less than the content in the corresponding normalized specimens combined with Figure 8 and Table 3.

**Table 3.** Texture composition of the decarburized samples, area ratio/%.

| Texture | Sample A | Sample B | Sample C |
|---|---|---|---|
| {110}<001> | 0.77 | 0.75 | 0.78 |
| {411}<148> | 0.75 | 0.72 | 0.73 |
| {111}<112> | 0.78 | 0.77 | 0.77 |

Goss texture, {411}<148> texture and {111}<112> texture contents show similarity in the specimens. But Figure 10 indicates that the specimen normalized at 950 °C has stronger {111}<112> texture. It means that 950 °C normalization annealed can make the texture {111}<112> has smaller deviation angle with the standard one, which is beneficial for Goss texture's abnormal growth in the secondary recrystallization process. In addition, the coincidence site lattice (CSL) theory suggests that $\sum 9$ grain boundaries around Goss texture has low energy and high mobility, which can promote Goss grain growth. According to the CSL theory, Figure 11 indicates 950 °C normalized specimen is better for decarburization annealing to obtain expected Goss texture. So, normalizing at 950 °C is recommended in the produce process of grain oriented silicon steel to form more fine low temperature inhibitors and obtain higher proportion and stronger Goss texture.

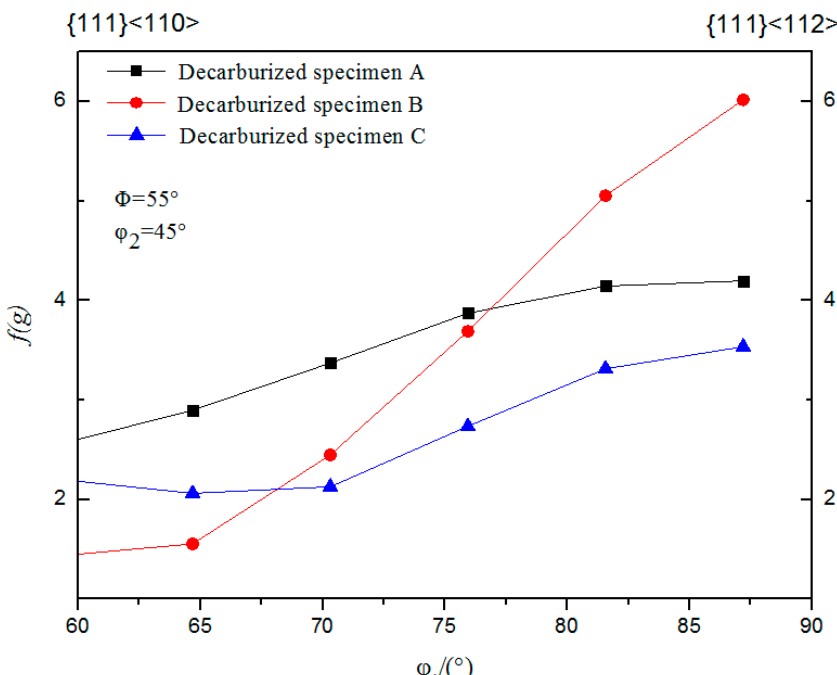

**Figure 10.** Variety of Gamma($\gamma$) texture in different specimens.

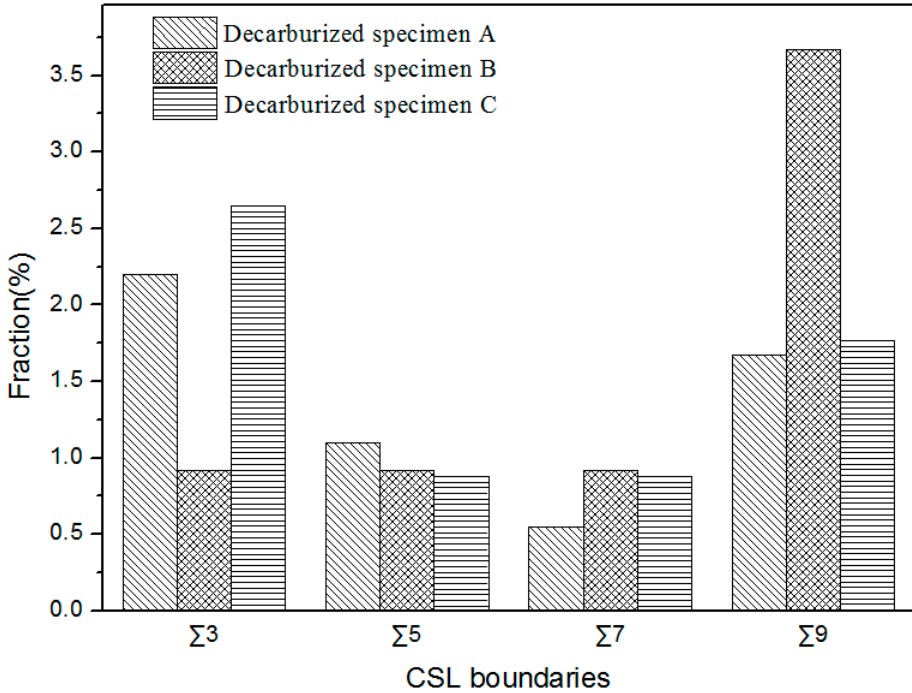

**Figure 11.** Variety of CSL grain boundaries in different specimens.

## 5. Conclusions

The evolution of inhibitors and the texture of primary recrystallization were investigated in Fe-3%Si steel bearing niobium and chromium. Conclusions are summarized as follows.

(1) Enough quantity and fine inhibitors with 20~40 nm in size are obtained in the tested specimens, mainly including of NbN, NbC, Nb(C,N), MnS, AlN, or their compounds, and only a little of $Cr_7C_3$ and CrN were found, which will ensure the inhibition effect during the secondary recrystallization process. The pining force of the inhibitors in the decarburized specimen has the relationship, $Z_{920°C} \approx 1.06Z_{950°C} \approx 2.26Z_{980°C}$, when considering the number and size of the precipitates.

(2) The content of Goss texture in the normalized specimens is 1.01%, 0.824%, 1.49%, and it decreases to 0.77%, 0.75% and 0.78%, respectively, in decarburized specimens. The specimen normalized at 950 °C has the minimum reduction (about 8.98%).

(3) After decarburization, the average grain size of the samples is 8.52 μm, 9.16 μm and 9.06 μm respectively, and the main textures {111}<112>, {411}<148> show little change, but the content of $\sum 9$ grain boundaries around the Goss grains is different. The specimen normalizing annealed at 950 °C and then decarburized has a maximum value 3.6%. The results show that the present normalization processes have no significant impact on the grain size of primary recrystallization, but it can affect texture distribution in the steels.

(4) The Goss grain obtained by normalizing annealed at 950 °C and then being decarburized is most likely to grow abnormally during the subsequent high temperature annealing process, and to ultimately obtain excellent properties.

**Author Contributions:** C.Z. and Y.L. (Yulong Liu) conceived and designed the experiments; Y.L. (Yulong Liu) and Y.W. performed the experiments; J.J. and Y.L. (Yulong Liu) analyzed the data; G.L. and Y.L. (Yu Liu) contributed analysis tools; Y.L. (Yulong Liu) wrote the first draft of the manuscript; C.Z. revised and approved the final version of the manuscript.

**Funding:** This research was funded by the National Natural Science Foundation of China (No. 51674180) and China Postdoctoral Science Foundation (No. 2013M540609).

**Acknowledgments:** The authors of this paper thank the National Natural Science Foundation of China and China Postdoctoral Science Foundation.

**Conflicts of Interest:** The authors declare no conflict of interest.

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
