# Peer review of "Effect of Normalizing Annealing Temperature on Precipitates and Texture of Nb-Cr-Bearing Decarburized Grain-Oriented Silicon Steels"

_metals, doi:10.3390/met9040457_

Round 1

Reviewer 1 Report

The paper presents an in-depth study into the effect of normalizing annealing temperature on precipitates and texture of Nb-Cr bearing decarburized grain oriented silicon steels. The paper is well-written and the results are either supported by existing literature or technical rationalisations.  

The authors should provide reasoning behind their methods. No rationale has been provided for choosing the temperatures, timing and dimensions of the ingots. Please clearly state whether they are based on past work of the authors or existing literature.

Some minor corrections:

Page 1; line 13: ……different temperatures. Enough…….

Page 1; line 42: ……Reports [14, 18] have shown that…..

Page 1; line 43: ……Li et al [15] obtained the…….

Page 12; line 178-179: There are no spaces between the words and the brackets.

Page 13; line 195: ……method [28] can be………

Page 13; line 204: ……which is a benefit……..

Page 13; line 213: ……Due to the fact that the strain…….

Page 13; line 224: ……could be explained……

Page 14; line 241: ……contents show similarity…….

Page 15; line 265: ……respectively, the……….

Author Response

Dear Reviewer:

Thank you for your comments concerning our manuscript entitled “Effect of Normalizing Annealing Temperature on Precipitates and Texture of Nb-Cr-Bearing Decarburized Grain-Oriented Silicon Steels” (ID: metals-480851). Those comments are all valuable and very helpful for revising and improving our paper, as well as the important guiding significance to our researches. We have studied comments carefully and have made correction which we hope meet with approval. Revised portion are marked in red in the paper. The main corrections in the paper are as flowing:

Comments and Suggestions 1:The authors should provide reasoning behind their methods. No rationale has been provided for choosing the temperatures, timing and dimensions of the ingots. Please clearly state whether they are based on past work of the authors or existing literature.

Responses 1: Thanks for your advices, all parameters are designed according to our simulating steelmaking composition of the samples in moderate scale experiments and industrial production practice. We have revised the manuscript in section 2 “2. Materials and Methods” as follows:

  Grain-oriented silicon steel specimens used in the present work were smelt in an 100 kg vacuum induction furnace. According to our simulating steelmaking composition of the samples in moderate scale experiments and industrial production experience, the smelting temperature was set at 1640℃, and the liquid steel was cast into 210 mm × 120 mm × (30~50) mm ingots in a fixed copper mold. The chemical composition of the specimens used in this study is listed in Table 1. Since the reheating temperature was about 1400℃ in the conventional grain-oriented silicon steel using MnS and AlN as inhibitors, the ingots in the present experiments were reheated at 1350 ℃ for 210 minutes to ensure inhibitors to dissolve completely. The temperatures of hot rolling process were set by referencing to the industrial production practice. The bloom rolling temperature was 1100 ℃ and the finishing rolling temperature was 1000 ℃. The slab was hot rolled to 2.3 mm and then modeling coiled at 570 ℃. Since Nb and Cr are used as assistant elements, two-stage normalizing annealing and single cold rolling with 90% reduction were followed. References[14, 15,18] have reported that the first stage normalizing annealing temperature varies from 1000 ℃ to 1120 ℃, timing from 1 minute to 6 minutes, and the second stage annealing temperature changes from 920 ℃ to 960 ℃, timing about 3 minutes. Theoretically, the annealing temperature can be set from 1150 ℃ to 850 ℃, which includes austenite changing to ferrite to obtain more and fine inhibitors. According to the industrial practice, the first stage annealing temperature was set at 1120 ℃ and kept 4 minutes. 

Comments and Suggestions2:Some minor corrections

Responses 2: Thanks for your advices, we have revised the related parts as follows:

1) Page 1; line 13” …… different temperature. Enough…….” were corrected as”……different temperatures. Enough…… ” (Page 1; line 13)

2) Page 1; line 42”.…..Reports[14,18] shown that……” were corrected as”.….. Reports[14,18] have shown that……”(Page 1; line 43)

3) Page 1; line 43:”……Li et al [15] got the……” were corrected as”……Li et al [15] obtained the…….”(Page 1; line 44)

4) Page 12; line 178-179: There are no spaces between the words and the brackets. The space was added. (Page 12; line 190-191)

5) Page 13; line 195:”……method [28] and be” were corrected as” ……method [28] can be………” (Page 13; line 208)

6) Page 13; line 204: “……which is benefit……”were corrected as” ……which is a benefit……..”(Page 13; line217)

7) Page 13; line 213: “……Due to the strain……”were corrected as”……Due to the fact that the strain…….”(Page 13; line 226)

8) Page 13; line 224:”……could explained……” were corrected as”……could be explained……” (Page 13; line 236)

9) Page 14; line 241:”……contents are similarity……”were corrected as” ……contents show similarity…….” (Page 14; line 254)

10) Page 15; line 265: “……respectively, and the……”were corrected as”……respectively, the……….” (Page 15; line 281 )

The authors’ revision:

(1) We have examined the manuscript carefully again, and all changed details have been marked in track changes mode in MS Word in the manuscript.

    (2) We have replace figure 2, figure 5, and figure 8, to keep all figures consistent.

Reviewer 2 Report

These EBSD results are not sufficient for analysis and it is necessary to provide X-ray data for reliable and justified conclusion.

Author Response

Dear Reviewer:

Thank you for your comments concerning our manuscript entitled “Effect of Normalizing Annealing Temperature on Precipitates and Texture of Nb-Cr-Bearing Decarburized Grain-Oriented Silicon Steels” (ID: metals-480851). We have studied comments carefullyWe want to make a further explain for you:

Comments and SuggestionsThese EBSD results are not sufficient for analysis and it is necessary to provide X-ray data for reliable and justified conclusion.

Responses to : We agree with your comments entirely. More macrostructure data can be obtained from X-ray analysis and the results are more accurate for analysis compared with EBSD as you mentioned. In our present work, the evolution of Goss texture in normalizing and decarburizing bands was investigated, which mainly originates from the subsurface of hot rolled band. We paid more attention on the microstructure of the samples and analyzed the longitudinal section of the specimens. In order to ensure the representative of the results, several observation fields were conducted in the experiments. The EBSD data were averaged on the basis of the statistical results.

On the other hand, the results such as coincidence site lattice (CSL) grain boundaries, grain size, especially for the variation of 9 grain boundaries around Goss texture, must be obtained from EBSD analysis results of orientation and morphology, while X-ray analysis could not provide these data. Hence, EBSD was chosen to analyze microstructure and texture in the present work. We replenished the reason for choosing EBSD as the analysis method in the manuscript as follows.

In the present work, the evolution of Goss texture in normalizing and decarburizing bands was investigated, which mainly originates from the subsurface of hot rolled band. The results such as coincidence site lattice (CSL) grain boundaries, grain size, especially for the variation of 9 grain boundaries around Goss texture are expected. So, EBSD was chosen to analyze microstructure and texture in the present work, which can provide orientation and morphology information. In order to ensure the representative of the results, several observation fields were conducted in the experiments. The EBSD results were averaged on the basis of the statistical results.

The authors’ revision:

(1) We have examined the manuscript carefully again, and all changed details have been marked in track changes mode in MS Word in the manuscript.

(2) We have replaced figure 2, figure 5 and figure 8, to keep all figures consistent.

Reviewer 3 Report

The paper reports a characterization of precipitates (size and texture) in grain-oriented silicon steels produced after annealing at 3 different temperatures. The paper is potentially of interest for Metals, but it has to be significantly improved to deserve publication. 

The first problem I encountered reading the paper is the poor level of English. Besides some odd phrasing (e.g. starting sentences with “Seen from”) the paper contains many long sentences and often it is hard to follow what the authors mean. Therefore I recommend extensive English revision.

As to the scientific side, my main concern is whether the differences among the three samples are really significant. This depends on the method used and the uncertainties associated to the results. Hwever, no result is reported with standard deviation and details of data processing are not reported. This leads to the following points:   

1- it looks like that the best annealing temperature is 950 °C, as it leads to lower Goss texture (0.8 % against 1.0 and 1.5% at other temperatures) and bigger grains (9.2 microns against 8.5 and 9.1). How is grain size calculated? From the text it looks like it was derived from eq 16, but it is not clear. Is there really a difference between the grain size at 980°C (9.1 um) and 950°C (9.2 um)?

 2 - In the conclusion, at point 2 the authors claim “Goss texture in decarburized specimens decreases in different degree, which is 0.77%, 0.75% and 0.78% “, while in the following point they observed that “the main texture {111} < 112>, {411} < 148> have little change”. Actually {411} < 148> changes from 0.72% to 0.75%, a gap of 0.03% as observed for Goss texture. From authors’ descriptions it looks like that the first change is significant, the second is not. In my opinion, none of them is significant.

3 - Line 90: “After decarbonized annealing, the content of carbon in the specimens decrease to about 0.0054 wt.%.” How was it calculated? Is reliable so small a difference?

4 - According to the experimental section the size of precipitates was derived by tem, I guess from images like those shown in figure 1. What is the resolution limit of the instrument? Is it really possible to resolve precipitates with a resolution of 10 nm? This is important as most of precipitates have the smallest size. It would be interesting to show in the figure examples of precipitates with different size.

To conclude, the goal of the paper, mentioned in the introduction, is:

“The optimized normalizing temperature is obtained. The results will elucidate the effect mechanism of annealing temperature on precipitation behaviors and texture evolution of Nb and Cr synergistically added grain-oriented silicon steels, which is expected to provide reference data for developing a proper annealing proposal. “

The optimum annealing temperature should be obtained by screening many different temperatures. With only 3 points it is not possible to see trends with temperature and this makes less reliable to define unambiguously which is the best temperature.

Other minor revisions:

- figure 1 should be split at least in 2 different figures. (the first up to panel h, the second with element mapping)

- last reference is written in red.

Author Response

Dear Reviewer:

Thank you for your comments concerning our manuscript entitled “Effect of Normalizing Annealing Temperature on Precipitates and Texture of Nb-Cr-Bearing Decarburized Grain-Oriented Silicon Steels” (ID: metals-480851). Those comments are all valuable and very helpful for revising and improving our paper, as well as the important guiding significance to our researches. We have studied comments carefully and have made correction which we hope meet with approval. Revised portion are marked in red in the paper. The main corrections in the paper are as flowing:

Comments and Suggestions 1The first problem I encountered reading the paper is the poor level of English. Besides some odd phrasing (e.g. starting sentences with “Seen from”) the paper contains many long sentences and often it is hard to follow what the authors mean. Therefore I recommend extensive English revision.

Responses 1: Thanks for your comment and suggestion. We tried our best to improve the manuscript and made some changes in the manuscript.

Comments and Suggestions 2: it looks like that the best annealing temperature is 950 °C, as it leads to lower Goss texture (0.8 % against 1.0 and 1.5% at other temperatures) and bigger grains (9.2 microns against 8.5 and 9.1). How is grain size calculated? From the text it looks like it was derived from eq 16, but it is not clear. Is there really a difference between the grain size at 980°C (9.1μm) and 950°C (9.2μm)?

Responses 2: We are sorry not to express the details of the calculation of grain size in Materials and Methods that The texture and grain size were quantitatively determined by HKL channel 5 electron back scatter diffraction (EBSD) system equipped at the field scanning electron microscope (FE-SEM).” We have revised the related part as following: “The microstructure and texture along the longitudinal section of the specimens were determined by the electron backscatter diffraction EBSDsystem equipped at a Nova 400 Nano field emission scanning electron microscope (FE-SEM). The texture content and grain size were calculate by the software HKL channel 5.” And in this paper, there is no difference between 980°C (9.1μm) and 950°C (9.2μm), we have revised the conclusion “The results show that the present normalization processes have little effect on the grain size of primary recrystallization….”, as “The results show that the present normalization processes have no significant impact on……”(Page 16; Conclusion 2)

Comments and Suggestions 3: In the conclusion, at point 2 the authors claim “Goss texture in decarburized specimens decreases in different degree, which is 0.77%, 0.75% and 0.78% “, while in the following point they observed that “the main texture {111} < 112>, {411} < 148> have little change”. Actually {411} < 148> changes from 0.72% to 0.75%, a gap of 0.03% as observed for Goss texture. From authors’ descriptions it looks like that the first change is significant, the second is not. In my opinion, none of them is significant.

Responses 3: We are sorry not to clearly express our observed results in Goss texture and {111} < 112>, {411} < 148>. Goss texture originated from hot rolling stage, we studied the “Goss seeds” changing in normalizing and decarburizing processes. The gaps of Goss texture are 0.24% (reduction from 1.01% in normalized specimen to 0.77% in decarburized specimen, and the other samples are the same), 0.07%, and 0.74%. {111} < 112> and {411} < 148> texture have effect on the Goss grain growth, and the gap 0.03% is the difference among the three decarburized specimens. Actually, the main texture {111} < 112>, {411} < 148> have little change. We have rewritten the related parts as following:

“(2) In decarburized specimens, the Goss texture content is 0.77%, 0.75%, 0.78% respectively, the main texture {111} < 112>, {411} < 148> in three specimens varies from 0.72% to 0.78%. And the average grain size of the samples is 8.52μm, 9.16μm and 9.06μm. The results show that the normalization temperature has no significant impact on the content of Goss, {111} < 112>, {411} < 148> and the grain size” 

(3) The content of Σ9 grain boundaries around the Goss grains is different in the specimens, the one normalizing annealed at 950 oC and then decarburized has the maximum value 3.6%.

Comments and Suggestions 4:  Line 90: “After decarbonized annealing, the content of carbon in the specimens decrease to about 0.0054 wt.%.” How was it calculated? Is reliable so small a difference?

Responses 4: We are sorry not to describe clearly how to get the result of carbon content “0.0054 wt.%.” We have revised in the section “Materials and Methods” and related part as following:

“About 100 connected fields were selected from each specimen. The CS-8800 Infrared Carbon Sulphur Analysis Meter was used to determine the carbon content in the decarburized specimens…

After decarbonized annealing, the content of carbon in the specimens was about 0.0054 wt.%.

Comments and Suggestions 5:  According to the experimental section the size of precipitates was derived by tem, I guess from images like those shown in figure 1. What is the resolution limit of the instrument? Is it really possible to resolve precipitates with a resolution of 10 nm? This is important as most of precipitates have the smallest size. It would be interesting to show in the figure examples of precipitates with different size.

Responses 5: The TEM used in the present work can magnify 300,000 times. But it is hard to focusing on the objective and make it clear. In order to make sure composition of the precipitates, 200,000 times were used to obtain element mapping, and the resolution is 20 nm. However, when 200,000 times was adopted, only several precipitates in the fields. So, we magnified 50,000~100,000 times to collect data to obtain number information by TEM.

We adopted the reviewer’s suggest and the different size of precipitates will be presented in corrected figure 2.

Comments and Suggestions 6:  The optimum annealing temperature should be obtained by screening many different temperatures. With only 3 points it is not possible to see trends with temperature and this makes less reliable to define unambiguously which is the best temperature.

Responses 6: Thanks for your advice, It is really true as the reviewer suggested that the optimum annealing temperature should be obtained by screening many different temperatures. There were much research on the normalizing process of grain oriented silicon steels. References[14, 15,18] have reported that the first stage normalizing annealing temperature varies from 1000 to 1120 , timing from 1 minute to 6 minutes, and the second stage annealing temperature changes from 920 to 960 , timing about 3 minutes. Theoretically, the annealing temperature can be set from 1150 to 850 , which includes austenite changing to ferrite to obtain more and fine inhibitors. According to the industrial production practice, the first stage annealing temperature was set at 1120 and kept 4 minutes, and the second stage temperature was set at 920, 950and 980.

Comments and Suggestions 7: figure 1 should be split at least in 2 different figures. (the first up to panel h, the second with element mapping)

Responses 7: Thank you very much for your good advice. We have made a correction according to the reviewer’s suggestion. And the Results were given in Figure 1 and Figure 2. And all figure numbers were rewritten.

Comments and Suggestions 8: last reference is written in red

Responses 8: We are sorry for our negligence of the last reference was written in red. And we have corrected it in the paper.

The authors’ revision:

(1) We have examined the manuscript carefully again, and all changed details have been marked in track changes mode in MS Word in the manuscript.

       (2) We have replace figure 2, figure 5, and figure 8, to keep all figures consistent.

Round 2

Reviewer 2 Report

Accept in present form

Reviewer 3 Report

I am positively surprised by the improvement of the new version. Now the manuscript is much easier to understand. Still the text contains some imperfections, but they can be fixed at a proofs stage.

e.g. line 266: " It means that 950 °C normalization annealed can make the texture {111}<112> has smaller deviation  angle with the standard one" sounds a bit weird. Also notice that line spacing changes on page 2.

As the authors addressed the requests and I suggest the paper to be accepted.